# Indian Healthcare Workers’ Issues, Challenges, and Coping Strategies during the COVID-19 Pandemic: A Cross-Sectional Study

**DOI:** 10.3390/ijerph20043661

**Published:** 2023-02-18

**Authors:** Anahita Ali, Santosh Kumar

**Affiliations:** Faculty of Public Health, Poornima University, Jaipur 303905, India

**Keywords:** COVID-19, healthcare workers, coping strategies, mental health

## Abstract

India faced the maximum number of mortalities and morbidities during the second wave of COVID-19. Healthcare workers (HCWs) worked in high-pressure and stressful environments. Therefore, this study aimed to assess the common issues, challenges, and coping strategies of HCWs, as well as the statistical association between demographical characteristics and coping strategies. A cross-sectional study was conducted with 759 HCWs, involving simple, random sampling in Rajasthan, India, between August 2022 and October 2022. Participants responded to a self-administered questionnaire that included a Brief–COPE inventory. The statistical association between commonly adopted coping strategies and demographic characteristics was tested using the chi-square test and Fisher’s exact test. A total of 669 (88%) respondents agreed that they faced issues during the COVID-19 pandemic: 721 (95%) participants experienced challenges at the personal level, 716 (94%) at the organizational level, and 557 (74%) at the societal level. Problem-focused coping strategies were frequently adopted by the participants. Gender, marital status, education, hours of work per day, and residential area were significantly associated with a problem-focused coping strategy (*p* < 0.05). The findings of this study reported a limited use of coping strategies by the participants during the public health crisis, despite facing issues and challenges at work. These findings highlight the need to assist HCWs in developing coping mechanisms to maintain good mental health at work.

## 1. Introduction

Public health crises may affect all aspects of human health, including physical health, mental health, and social well-being. Historically, outbreaks, epidemics, and pandemics all made an impact on the people who were affected in some way. Coronavirus disease 2019 (COVID-19) is one such crisis that caused mixed feelings of anxiety, fear, and loneliness. Keeping in mind the positive side of this crisis, the response toward the pandemic may have helped people cope better with the situation. Since 2020, healthcare workers (HCWs) worldwide have played an important role in managing the pandemic. However, the impact of the pandemic on their well-being has drastically altered their health status worldwide [1].

Since the onset of the pandemic, HCWs have globally faced mental health issues such as anxiety, depression, sleeplessness, and stress [2]. Uncertainty, constantly changing guidelines for professional practices, working long shifts, dealing with patients, distant family and friends, shortage of medical personnel and medical resources such as protective equipment (PPE) are only a few of the problems reported. Studies have suggested that negative workplace experiences can cause moral harm, burnout, and ethical dilemmas [3]. In addition, demographic characteristics may act as risk factors for poor mental health during the COVID-19 pandemic. For instance, medical professionals who were not married were more vulnerable to psychiatric disorders than married professionals [3,4]. A similar study reported that female HCWs who feared contracting COVID-19 and watching COVID-19 related news were more likely to develop stress, anxiety, and depressive symptoms [5]. Globally, commonly reported challenges for HCWs include having been subjected to immense psychological suffering and physical stress, incompetence at work, fear of contracting a COVID-19 infection, ethical dilemmas, sleep-related problems, and difficulty communicating their problems to others [6,7]. 

During the second wave of COVID-19, India faced the highest number of mortalities and morbidities. Indian HCWs worked in high-pressure and stressful environments, due to which those working in COVID-19-designated hospitals reported clinically significant depression and poor sleep [5]. Feeling confined, having difficulty breathing, and experiencing frustration were some of the challenges reported by Indian nurses after wearing personal protective equipment (PPE) kits at work [8]. Doctors posted in a COVID-19-dedicated ward at an Indian tertiary hospital very often felt nervous, upset, and were unable to cope with the COVID-19 situation [9]. Studies reported severe to moderate levels of anxiety among Indian HCWs in 2020 [10]. Symptoms of stress, anxiety, and poor sleep quality were most commonly reported by Indian HCWs [5,11]. 

Coping plays an important role maintaining good mental health while dealing with stressful situations in everyday life and even with public health crises such as the COVID-19 pandemic. Coping is defined as “constantly changing cognitive, and behavioral efforts to handle specific external, and/or internal requirements assessed as taxing or exceeding the person’s resources” [12]. Individuals deal with stress in unique and different ways; some may prefer to communicate emotions openly, whereas others may actively reconsider the source of the distress and its occurrence so that it seems less stressful [13]. Psychologists have categorized coping strategies into two strategies: namely, emotion-focused and problem-focused strategies. Individuals generally prefer avoidant coping to minimize emotional stress instead of coping with it. Problem-focused coping involves identifying means of dealing with a problematic event or situation, such as using hand sanitizers to reduce the risk of contracting COVID-19. 

Emotion-focused coping is the most commonly adopted coping strategy for managing COVID-19 [14]. Interestingly, healthcare personnel were engaged in their desire to help the impoverished during the COVID-19 pandemic with a commitment to maintain critical services, which is meaning-focused coping strategy [14]. As part of problem-focused coping, HCWs attempted to realign their services, use PPE, practice effective cleaning methods, and follow patient safety measures [15]. A recent, systematic analysis of the impact of COVID-19 on the mental health of HCWs found that a lack of social support, poor communication, maladaptive coping, and a lack of training were risk factors for developing psychiatric problems [4].

In addition to simply adopting a coping strategy, the availability and accessibility of resources (commonly known as coping resources) are critical for coping with a stressful situation. For instance, all “physical, infrastructural, and human capital entities” are referred to as resources. Unfortunately, challenges associated with resource availability increased during the COVID-19 pandemic. HCWs reported various issues related to resource availability, allocation, and adequacy. Coping resources are essential, even for the general population. For instance, people visit gardens and yoga centers near their houses for recreational activities [16,17]. However, during a public crisis such as COVID-19, limited access to coping resources may influence the coping behavior of individuals. For instance, limiting the frequency of visits to gardens or other green spaces may affect coping behavior and, in turn, worsen mental health. Therefore, it was recommended that green spaces be made accessible for mental health purposes during the COVID-19 pandemic [18]. 

Altogether, the negative impact of the COVID-19 pandemic on healthcare systems, including HCWs, has made it mandatory to assess the mental health status of HCWs working during the emergency and its potential consequence on their well-being. This pandemic is likely to continue to exist and affect HCWs. Disasters such as COVID-19 may act as independent risk factor for poor mental health among HCWs [19]. Therefore, it is important to ensure the complete well-being of HCWs by focusing on all aspects of their health. In this context, data on the impact of COVID-19 on mental health are needed to call for actions. This study aimed to assess the common issues, challenges, and coping strategies of Indian HCWs during the COVID-19 pandemic. 

### Aims and Objectives of the Study

As the overall health and well-being of HCWs are important for a healthy healthcare system, we aimed to analyze the issues, challenges, and coping strategies of HCWs during the COVID-19 pandemic in Rajasthan, India. The specific objectives of this study were as follows:To identify issues and challenges faced by HCWs during the second wave of the COVID-19 pandemic in Rajasthan;To assess the coping strategies of HCWs during the second wave of the COVID-19 pandemic in Rajasthan;To assess the statistical associations between demographic characteristics and coping strategies.

## 2. Materials and Methods

### 2.1. Type of Study 

A cross-sectional, observational study was conducted in the Rajasthan Government District Hospital in India from August to October 2022. Rajasthan is the largest state in India in terms of geographical area and accounts for 5% of the country’s population. The state is divided into six zones and thirty-three districts. Every district has its own healthcare facilities that provide primary, secondary, and tertiary care. Some large districts also have satellite hospitals that serve as extensions to these health facilities. The total number of HCWs working in the target district hospital was 3000. 

### 2.2. Study Participants and Sample 

All the HCWs (doctors, nurses, allied workers, and administrative workers) were considered for sampling. The sample size for the study was calculated using Cochran’s formula [20]. 

At a 95% confidence interval and a 5% acceptable margin of error, we arrived at a sample size of 764 (plus a 10% non-response rate), of which the sample size for doctors was 245, the sample size of nurses was 307, the sample size of allied workers was 194 (Table A1). As the administrative staff were small in number (18 employees), we included all of them. A total of 840 HCWs were taken into consideration and contacted, out of which 81 incomplete questionnaires were obtained. As a result, a final total sample of 759 HCWs were included in this study. Doctors, nurses, allied workers, and administrative staff, working in different departments of the hospital, were included in the study. Other nonmedical personnel (such as security guards or receptionists) employed in the hospital were excluded. Those who could not be approached after three attempts were also excluded.

The participants were selected via a simple, random sampling method in which every participant was assigned a unique number. A list of participants was created, and the participants were selected randomly. 

### 2.3. Data Collection Instrument

A self-administered questionnaire, developed by the researchers, was used to collect information about the participants. The questionnaire was developed in both English and the national language (Hindi). The questionnaire was divided into three sections (as mentioned below). 

#### 2.3.1. Demographic Characteristics 

The first section of the questionnaire assessed demographic characteristics (age, sex, education, marital status, work experience, type of shift, working hours, living status, type of accommodation, and distance from green spaces). 

#### 2.3.2. Issues and Challenges 

The second section addressed common issues and challenges such as “challenges in work life since COVID-19 s wave?” and “Challenges faced due to stress at the organizational, individual, and societal levels” [21,22,23]. The Cronbach’s alpha for this section of the questionnaire was 0.66. 

#### 2.3.3. Coping Strategies 

The respondents’ coping styles were examined using the Brief–COPE Inventory, which is a condensed version of Carver’s COPE scale [22,24]. The inventory was used to study three types of coping strategies on the Likert scale: namely, emotion-focused, problem-focused, and meaning-focused strategies. The coping styles were measured using statements on a four-point scale (1 = I haven’t been doing this at all, 2 = I have been doing this a little bit, 3 = I have been doing this moderately, and 4 = I have been doing this a lot). The data collection tool was employed in both English and Hindi. 

The psychometric parameters of the Brief–COPE questionnaire were established with 37% of the total variance in a recent study in 2021 that verified the psychometric qualities of the Brief–COPE assessment to ensure its use by nursing staff in the UAE [23,25]. Among Indian patients with HIV/AIDS, the Brief–COPE in Tamil yielded a five-factor (17-item) model that accounted for 41.5% of the overall variance [24,26]. Carver reported and established the validity and reliability of the Brief–COPE scale in the original scale, with a Cronbach’s alpha of 0.57–0.90 [25,27].

### 2.4. Ethical Consideration

This study was approved by the institutional review board and ethics committee of the district hospital (SNMC/IEC/2022/618). All participants provided written informed consent for voluntary participation. The participants were assured of the confidentiality and non-disclosure of their personal identities. The researchers followed all procedures in accordance with the ethical standards of the IRB (JSPH-IRB/2022/06/26) and as outlined in the Declaration of Helsinki. 

### 2.5. Data Analysis 

The collected data were presented as descriptive statistics using the Epi-info software, version 7. The mean scores and standard deviation were evaluated at a 95% confidence interval for the coping strategies. Commonly reported issues and challenges at different levels were compared with various demographic characteristics of the participants using the chi-square test and Fisher’s exact test, where appropriate. The statistical associations of commonly adopted coping strategies with various demographic characteristics of the participants were tested using the chi-square test and Fisher’s exact test, where appropriate. A *p* < 0.05 was considered to indicate the statistical significance of the variables.

## 3. Results

### 3.1. Demographic Characteristics

Data from a total sample of 759 HCWs (after removing 81 incomplete questionnaires) were analyzed in this study. Table 1 presents the demographic characteristics. 

### 3.2. Issues and Challenges

Of the participants, 669 (88%; Mean = 0.9 and SD = 0.01) reported issues they faced for several days during the second wave of the COVID-19 pandemic, such as having little interest in working (*n* = 139; 18%), not being able to stop or control their worrying (*n* = 164, 22%), feeling down, depressed, and hopeless (*n* = 395; 52%), and feeling nervous or anxious (*n* = 273, 36%). 

The participants reported the challenges they faced since the second wave of the COVID-19 pandemic. At the personal level (Mean = 1.02, SD = 0.07), out of 759 participants, 395 (52%) felt emotionally tired, 330 (44%) were stressed when exposed to new COVID-19 infected patients, 89 (11%) felt incompetent at work, 525 (69%) were afraid of contracting a COVID-19 infection, 218 (29%) faced ethical dilemmas, 222 (29%) had sleep-related problems, and 67 (9%) faced difficulty in communicating their problems to others. 

At the organizational level (Mean = 1.03; SD = 0.08), 522 (69%), out of 759 participants reported a shortage of PPE, 604 (80%) had to wear protective equipment every day, 61 (8%) reported a lack of organizational support at the hospital, 561 (74%) reported that the COVID-19-related guidelines were unclear, 71 (9%) said they were forced to work overtime, 133 (18%) reported poor organizational preparedness for any disaster, 135 (18%) reported a shortage of mental health support services, and 383 (51%) reported a lack of incentives provided to them. 

At the societal level (Mean = 1.3; SD = 0.01), 380 (50%), out of 759 participants experienced a fear of alienation from society, 371 (49%) faced issues such as poor support from society, 126 (17%) faced low social acceptance, and 312 (41%) faced stigma and discrimination from society.

### 3.3. Coping Strategies 

The participants reported three types of coping strategies, as shown in Table 2 below. Only the problem-focused coping strategy was adopted for most of the days, while other two were used moderately or for some days. 

A total of 449 (59%, Mean = 1.2; SD = 0.01) participants reported little or limited use of an emotion-focused coping strategy. Among these, 293 (65%) received emotional support from others, 66 (17%) suppressed their unpleasant feelings in order to suppress their emotions, and 38 (8%) criticized themselves. However, a few emotion-focused coping strategies were adopted every day by the participants. A total of 16 (4%) participants felt uncomfortable because of their issues, and 26 (6%) accepted the reality that they were facing those issues (issues and challenges related to poor mental health); only 10 (2%) participants expressed their negative feelings relating to the issues and challenges to others, and 500 (75%) participants watched TV, read books, or slept during their leisure time to distract themselves from their issues and challenges.

A total of 68 (9%, Mean = 2.1; SD = 0.02) participants adopted problem-focused coping strategies for most of the days. Among these, 15 (22%) participants attempted to take some action to make their mental health better, 25 (38%) had received advice or assistance from other people, 16 (24%) participants tried to view their issues in a different light, to make it seem more positive, and 12 (18%) participants attempted to generate a strategy for addressing their issues and challenges during COVID-19. 

A total of 668 (88%, Mean = 1.6; SD = 0.02) participants reported little or limited use of meaning-focused coping strategies. Of this number, 68 (10%) participants used alcohol or other drugs to feel better every day, 25 (4%) turned to work or other activities to distract themselves from their issues and challenges, and 75 (11%) performed religious practices such as praying or worshiping. 

As is shown in Table 3 below, the *p*-values obtained from the chi-square test and the Kruskal–Wallis H test showed a significant association between the demographic characteristics and the coping strategies adopted by the participants. Gender, marital status, shift type, hours of work every day, residential area, type of accommodation, and distance from green spaces were statistically associated with emotion-focused coping strategies (*p* < 0.05). Gender, marital status, education, hours of work every day, and residential area were significantly associated with problem-focused coping strategies (*p* < 0.05). Marital status, experience, living status, shift type, hours of work every day, residential area, and distance from green spaces were statistically associated with meaning-focused coping strategies (*p* < 0.05). Hence, a few of the demographic characteristics of the HCWs were statistically associated with an increased adoption of coping strategies.

## 4. Discussion

The majority of participants in this study faced issues such as having little interest in working, worrying, feeling down, depressed, and hopelessness for several days during the pandemic. The most commonly reported challenges at the organizational level included a shortage of PPE and wearing PPE every day, which led to increased levels of physical burdens, stress, tiredness, and prolonged hours of work without eating or drinking. In response to these issues and challenges, the participants adopted problem-focused coping strategies on most days during the pandemic. 

### 4.1. Issues and Challenges 

Previous studies reported severe-to-moderate levels of anxiety among Indian HCWs in 2020 [10], which is consistent with the current results. However, the results of this study varied between the sexes and ages of the participants [28,29]. Symptoms of stress, anxiety, and poor sleep quality were reported by Indian HCWs, which agree with the findings of this study [5,11]. The participants of this study reported feeling down, anxious, and hopeless on a scale of several days to most days as the most prevalent problems among HCWs [30]. 

Similar to previous findings, this study reported a significant association between demographic variables for HCWs such as their education level, experience level, the distance of green spaces from their home, and the frequency of visiting green spaces for recreational activities, with *p* < 0.05 [19]. It is likely that experienced HCWs or senior staff faced more issues than their junior staff because, due to their senior designation, the workload increased in the hospital. Overall, a disaster such as COVID-19 may act as an independent risk factor for increased stress among HCWs [19]. 

Most participants in this study reported various challenges that they faced at the personal, organizational, and societal levels. At the personal level, up to half of the participants said that they felt emotional tiredness or burnout; this was also reported by hospital managers in Pakistan [6]. This study suggests that increased challenges for HCWs in their personal life may result from an increased workload at the workplace, the designation of HCWs, their living status, the distance of green spaces from their home, and the frequency with which they visit green spaces for recreational activities [9]. Incompetence at work, the fear of contracting a COVID-19 infection, ethical dilemmas, sleep-related problems, and a difficulty in communicating their problems to others were commonly reported challenges faced by the participants, similar to previous studies across countries [6,7,31]. It is likely that being involved directly with COVID-19-positive patients causes more stress and fear among HCWs [32]. 

At the organizational level, participants reported challenges such as a lack of PPE, wearing masks, protective measure kits every day, and a lack of incentive. Most studies reported similar challenges associated with a high prevalence of anxiety and depressive symptoms among HCWs [33]. As reported in this study, improper guidelines and a lack of complete information increased the challenges of working in the hospital with heavy PPE worn by HCWs posted in critical care areas such as intensive care units and operation theatres [33,34]. 

Participants in this study reported challenges faced at the societal level such as boycotts from their community, myths and rumors about HCWs, a lack of support, and alienation from their lessor [34,35]. Those living alone and far from their families faced more challenges and lacked support from the society they lived in [35,36]. Interestingly, this study did not find a significant association between participants’ marital status and the level of challenge they faced. These findings contrast with those of HCWs from another COVID-19-designated hospital in India [5].

### 4.2. Coping Strategies

This study reported little or no use of meaning-focused coping strategies to cope with the COVID-19-related issues by most participants (*n* = 668), whereas emotion-focused and problem-focused coping strategies were used by a few participants (*n* = 449 and *n* = 409, respectively). 

#### 4.2.1. Emotion-Focused Coping

Emotional imbalance is characterized by an unprecedented number of emotions, anxiety, and the suppression of feelings [36,37]. Emotion-focused coping was the most commonly adopted coping technique for dealing with the COVID-19 threat [14]. However, the participants in this study reported a moderate use (*n* = 248) of emotion-focused coping, which was statistically associated with sex. Indulging in hobbies and spending time with family were the most-cited methods of emotional regulation [14]. However, this is contrary to the results of this study, in which only a few participants regulated their emotions in this way (*n* = 74). Peer support is viewed as a source of encouragement and connection that delivers various positive emotions during times of difficulty and significantly reduces stress [7,38,39,40]. Nevertheless, these findings contradict the observations of this study, in which emotion-focused strategies were not found to be associated with peer support or good relationships at work. Praying together, working together, reminding each other about infection control, listening to each other, and comforting each other are some documented methods of peer support [14]. These results agree with those of this study, with many participants (*n* = 293) stating that they received peer support on most days. 

#### 4.2.2. Problem-Focused Coping 

During the pandemic, healthcare personnel were engaged in their desire to help the impoverished and had a commitment to maintain critical services [14]. As part of problem-focused coping, HCWs attempted to realign their services, use personal protective equipment, practice effective cleaning methods, and follow patient safety measures [15] that were statistically associated with their designation and experience level. The participants in this study reported little use of problem-focused coping (*n* = 409), and only 68 participants reported using this coping strategy on most days. In contrast, HCWs reported the use of PPE and disinfection methods to approach this problem. 

#### 4.2.3. Meaning-Focused Coping 

Spirituality was mentioned as an important coping strategy, such as reciting morning prayers and attending weekly spiritual events [15]. Similar behaviors were reported by the participants in this study, with little-to-moderate use of spirituality (*n* = 668). In the current study, HCWs focused on their moral values, such as serving the community and providing social services (*n* = 106) for most of the days. These findings contradict those of other groups from more developed economies such as China, Canada, and the USA [12,41]. Overall, the assumption regarding the significant association between demographic characteristics and coping strategies supports the alternate hypothesis based on the significant *p*-values obtained, as depicted in Figure 1. 

Figure 1 below illustrates the adoption of the transactional model for stress and coping in the current scenario. Some HCWs felt a high intensity of the stressor (COVID-19), while others were not disturbed by it. The effect of COVID-19 on HCWs led to the adoption of primary appraisal. Few HCWs continued to feel that their lives were in danger, while others felt that they were safe (due to personal protective equipment) and not as vulnerable to the virus as others. Possible reasons for these variations in perceived susceptibility and severity were the sex, marital status, designation, shift type, shift time, and experience level of the HCWs. These demographic factors act as risk factors and add stress to the situation. Mutual stress, caused by the pandemic and demographic factors, led to the adoption of coping strategies to reduce overall stress. HCWs with sufficient coping resources for handling a stressor and those who did not feel any challenge in coping adopted a positively engaged coping strategy; however, HCWs who faced challenges in coping with the situation adopted a negative coping style, such as the avoidance or denial of the situation. The coping outcomes included an enhanced emotional well-being, positive changes in health behavior, and positive changes in the health status of HCWs during the COVID-19 pandemic.

This study, conducted among HCWs of a government district hospital who coped with various issues and challenges during the second wave of the COVID-19 pandemic, is the first of its kind. However, this study had certain limitations. First, the sample size may not provide generalizable results that can be applied to all HCWs across regions. Second, this was a cross-sectional study. The coping behavior of HCWs may change with time as they face changing situations every day, although the issues and challenges may have lessened after the first wave of the pandemic (this study may not have fully captured their true problems). Changing coping behaviors should be addressed in longitudinal and cross-sectional studies. Third, because a self-administered questionnaire was used to collect data, the estimated associations and obtained results might have a reporting bias as the participants might have reported or revealed selective information. Finally, although HCWs from different departments were included, differences between HCWs from different departments were not analyzed. Therefore, the risk factors associated with mental health impact of the COVID-19 pandemic across different departments of the hospitals should be examined in future studies.

## 5. Conclusions

This study reported the limited use of coping strategies by HCWs during a public health crisis despite facing issues such as having little interest in performing work at the hospital; worrying; feeling down, depressed, hopeless; and experiencing challenges at the personal; organizational; and societal levels while at work. The maximum number of participants focused on managing their negative emotions with the help of watching TV, reading books, and sleeping. HCWs working in healthcare settings are more prone to developing mental-health-related issues and challenges [42] which, in turn, require the adoption of more coping strategies. The statistically significant association obtained in this study suggests that spending more time in the hospital than usual influences coping behaviors and, in turn, the adoption of coping strategies. However, the limited use of coping strategies by Indian HCWs indicates the need for assisting them in developing coping mechanisms for maintaining good mental health at work. We suggest that future studies should explore different factors, such as mediating factors that affect the adoption of coping strategies, through mediation model analysis.

## Figures and Tables

**Figure 1 ijerph-20-03661-f001:**
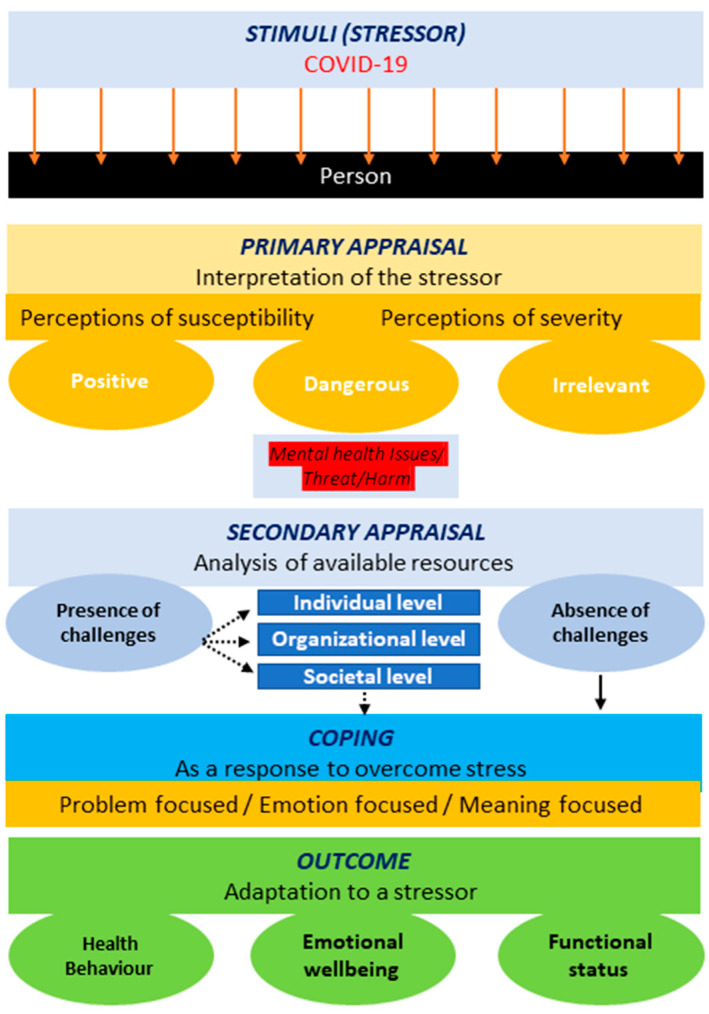
Adapted transactional model of stress and coping [28].

**Table 1 ijerph-20-03661-t001:** Participant demographic characteristics.

Demographic Characteristic	Category	Frequency (%)
Healthcare workers	Doctors	270 (36%)
	Nurses	325 (43%)
	Allied workers	146 (19%)
	Administrative staff	18 (2%)
Gender	Male	375 (49%)
	Female	384 (51%)
Age group	20–24 years	28 (4%)
	25–44 years	584 (77%)
	45–64 years	147 (20%)
Education	Diploma	268 (35%)
	Graduate	286 (38%)
	Post-graduate	197 (26%)
	Super specialty	8 (1%)
Marital status	Single	179 (24%)
	Married	562 (74%)
	Divorced	4 (0.5%)
	Separated	1 (0.1%)
	Widow	13 (1%)
Years of experience in the current hospital	1–5 years	450 (59%)
	6–10 years	157 (21%)
	More than 10 years	152 (20%)
Shift type	Daytime	476 (63%)
	Nighttime	74 (10%)
	Rotation	209 (27%)
Hours of work every day	Up to 6 h	462 (60%)
	Up to 8 h	243 (32%)
	Up to 12 h	54 (7%)
Living status	Alone	104 (14%)
	With family	600 (80%)
	With friends	34 (5%)
	With colleagues	21 (3%)
Type of accommodation	Staff quarters	114 (15%)
	Owned house	437 (58%)
	Rented house	198 (26%)
Distance of green spaces from accommodation	Within 0.5 km	285 (38%)
	Between 0.5 and 2 km	234 (31%)
	More than 2 km	240 (32%)

**Table 2 ijerph-20-03661-t002:** Summary of coping strategies reported by the participants.

Coping Style	Frequency: Used a Little Bit (%)	Frequency: Used Moderately (%)	Frequency: Used Most Days (%)
Emotion-focused coping	449 (59%)	248 (33%)	-
Problem-focused coping	409 (54%)	280 (37%)	68 (9%)
Meaning-focused coping	668 (88%)	89 (12%)	-

**Table 3 ijerph-20-03661-t003:** Means scores of coping strategies reported by the participants according to demographic characteristics.

Coping Strategy	Emotion-Focused Coping	Meaning-Focused Coping	Problem-Focused Coping
	Mean (SD *)	Mean (SD *)	Mean (SD *)
Current designation/role			
Doctors	1.9 (0.3)	1.6 (0.3)	1.9 (0.5)
Nurses	1.9 (0.4)	1.6 (0.4)	2.2 (0.5)
Allied workers	2.0 (0.5)	1.7 (0.4)	2.2 (0.7)
Administrative staff	2.0 (0.5)	2.6 (0.7)	2.6 (0.7)
*p* ***	0.072	<0.001	<0.001
Age			
20–24	2.5 (0.3)	1.4 (0.4)	2.5 (0.2)
25–44	2.2 (0.6)	1.8 (0.5)	2.3 (0.7)
45–64	2.2 (0.3)	1.6 (0.3)	2.4 (0.2)
*p* ***	<0.001	<0.001	<0.001
Gender			
Male	1.9 (0.4)	1.6 (0.4)	2.1 (0.6)
Female	1.9 (0.3)	1.5 (0.3)	2.1 (0.6)
*p* ***	0.012	<0.001	<0.001
Education			
Diploma	2.2 (0.4)	1.4 (0.0)	2.0 (0.3)
Graduate	1.9 (0.4)	1.6 (0.4)	2.0 (0.5)
Post-graduate	1.8 (0.4)	1.4 (0.4)	1.9 (0.5)
Super specialty	2.0 (0.4)	1.7 (0.4)	2.4 (0.7)
*p* **	0.091	<0.001	<0.001
Marital status			
Single	1.9 (0.4)	1.5 (0.4)	2.0 (0.5)
Married	2.6 (0.2)	1.6 (0.4)	2.1 (0.6)
Divorced	1.8 (0.1)	1.9 (0.2)	3.0 (0.1)
Separated	1.9 (0.3)	1.4 (0.3)	1.5 (0.1)
Widow	1.9 (0.3)	1.6 (0.3)	2.0 (0.6)
*p* ***	0.033	0.051	<0.001
Years of experience in the current hospital			
1–5 years	1.9 (0.4)	0.2 (0.5)	2.0 (0.5)
6–10 years	1.9 (0.4)	0.1 (0.6)	2.1 (0.6)
More than 10 years	1.9 (0.4)	0.2 (0.5)	2.1 (0.5)
*p* ***	0.091	<0.001	<0.001
Shift type			
Daytime	1.8 (0.4)	1.5 (0.4)	2.1 (0.6)
Nighttime	2.0 (0.5)	1.7 (0.5)	1.8 (0.4)
Rotation	2.0 (0.3)	1.7 (0.5)	2.2 (0.5)
*p* **	<0.001	0.010	<0.001
Hours of work every day			
Up to 6 h	1.9 (0.4)	1.6 (0.4)	2.0 (0.6)
Up to 8 h	1.9 (0.4)	1.6 (0.4)	2.1 (0.6)
Up to 12 h	2.0 (0.3)	1.5 (0.3)	2.2 (0.4)
*p* **	0.051	0.015	0.010
Living status			
Alone	1.8 (0.3)	1.5 (0.4)	2.1 (0.5)
With family	1.9 (0.4)	1.6 (0.4)	2.1 (0.6)
With friends	1.8 (0.4)	1.5 (0.2)	2.1 (0.5)
With colleagues	1.8 (0.2)	1.5 (0.5)	2.0 (0.5)
*p* **	<0.001	<0.001	0.010
Type of residential area			
Rural	2.3 (0.5)	2.0 (0.6)	2.0 (0.4)
Semi-urban	1.9 (0.4)	1.6 (0.4)	1.8 (0.4)
Urban	1.9 (0.4)	1.6 (0.4)	2.1 (0.6)
*p* ***	<0.001	0.028	<0.001
Type of accommodation			
Staff quarters	2.0 (0.4)	1.6 (0.4)	2.3 (0.6)
Owned house	1.9 (0.4)	1.6 (0.3)	2.0 (0.5)
Rented house	2.0 (0.4)	1.6 (0.5)	2.2 (0.6)
*p* ***	0.010	0.137	<0.001
Distance of green spaces from accommodation			
Within 0.5 km	1.9 (0.3)	1.6 (0.4)	2.1 (0.6)
Between 0.5 and 2 km	1.9 (0.4)	1.6 (0.3)	2.1 (0.5)
More than 2 km	1.9 (0.3)	1.6 (0.5)	2.0 (0.6)
*p* ***	<0.001	0.331	0.010
Frequency of visiting green spaces during second wave of COVID-19			
Everyday	1.7 (0.3)	1.4 (0.3)	1.5 (0.3)
Most of days	2.0 (0.4)	1.4 (0.3)	1.8 (0.6)
Once a week	2.0 (0.4)	1.6 (0.4)	2.1 (0.5)
Once a month	1.9 (0.4)	1.6 (0.4)	2.3 (0.7)
Never	1.9 (0.3)	1.6 (0.5)	2.1 (0.4)
*p* **	<0.001	<0.001	<0.001

* Standard deviation, ** Chi-square test, *** Kruskal Wallis H test.

## Data Availability

Data presented in this study are available upon request from the corresponding author.

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
