# Peer review of "Indian Healthcare Workers’ Issues, Challenges, and Coping Strategies during the COVID-19 Pandemic: A Cross-Sectional Study"

_ijerph, 2023, doi:10.3390/ijerph20043661_

Round 1

Reviewer 1 Report

This cross-sectional study explores issues, challenges, and coping strategies in healthcare workers during the second wave of the COVID-19 pandemic in India. A good study design was chosen to explain the stated objectives. Even in the abstract, a clear conclusion of the study based on the obtained results is missing. I believe that the presentation of even 5 P values that describe the association of certain demographic characteristics with directional wearing is unnecessary, and should be combined into one p-value.

Methods: It is not necessary to show the formula of how to calculate the sample size, but a reference can be added in the text. The approval number of the Ethics Committee is missing.

Results: Too many percentages are listed in the text, and the results are very difficult to read. Demographic characteristics can be written in table 2 next to the corresponding characteristic for better visibility of the results. Statistical data processing is very superficial. Most of the data was processed through percentages without a more detailed analysis of the relationship. In Table 2, how can the P value be = 0?? Also, P values should have an equal number of decimal places. Some of the results presented in part 3.3. is quite difficult to follow. Why is the description for Figure 1 inserted in the results? I think that this kind of description has no place in the results, especially because it describes and comments on Figure 1, which has no direct connection with the presentation of the obtained results.

Discussion: The first sentence of the discussion should summarize the key results obtained in the study, and not present a description of the study and general assumptions. Apart from the comment that the study cannot be generalized due to the sample size(?), no limitations of the conducted study were presented. I believe that the listed items are major omissions that must be corrected.

Conclusion: Too long, unclear and does not present a concrete conclusion of the study.

None of the references are written correctly according to the journal's guidelines (all are written in the usual Vancouver style).

In its current form, I believe that the paper should not be accepted for publication because the quality of its content and the results and conclusions obtained are not up to the standard of the journal to which it was submitted. This paper needs better data analysis and changes in the discussion area. 

Author Response

Dear reviewer, 

Greetings

We thank you for your time and effort in reviewing our manuscript. The feedback has been invaluable in improving the content and presentation of the paper. In accordance with your comments, we have revised our manuscript. The changes along with English editing are highlighted in the attached manuscript, and our point-by-point responses are given in "red" after the word “Response” below-

Point1: This cross-sectional study explores issues, challenges, and coping strategies in healthcare workers during the second wave of the COVID-19 pandemic in India. A good study design was chosen to explain the stated objectives. Even in the abstract, a clear conclusion of the study based on the obtained results is missing. I believe that the presentation of even 5 P values that describe the association of certain demographic characteristics with directional wearing is unnecessary, and should be combined into one p-value.

Response1: A clear conclusion is added in the abstract. Yes, we agree that the p-values must be combined into one p-value and therefore, we have done the changes in the p-values. Now they are mentioned as (p<0.05). 

Point2: Methods: It is not necessary to show the formula of how to calculate the sample size, but a reference can be added in the text. The approval number of the Ethics Committee is missing.

Response2: As per your suggestion, we have removed the formula and have given the reference number for the same. We have also mentioned the approval number of EC and IRB approval that we received for this study. 

Point3: Results: Too many percentages are listed in the text, and the results are very difficult to read. Demographic characteristics can be written in table 2 next to the corresponding characteristic for better visibility of the results. Statistical data processing is very superficial. Most of the data was processed through percentages without a more detailed analysis of the relationship. In Table 2, how can the P value be = 0?? Also, P values should have an equal number of decimal places. Some of the results presented in part 3.3. is quite difficult to follow. Why is the description for Figure 1 inserted in the results? I think that this kind of description has no place in the results, especially because it describes and comments on Figure 1, which has no direct connection with the presentation of the obtained results.

Response3: Yes, we agree that there were too many numbers and percentages to read. Therefore, we have created a separate demographic characteristic table for better understanding and clarity. As the main aim of this study was to assess the coping strategies and statistical associations, we had performed simple statistical analysis. Also, we believe that the study design (a descriptive study) supports this. Yes, we agree that a detailed analysis is expected, however this study is a part of PhD thesis. The main study proposal also has qualitative study design (for data triangulation) and mediation analysis of the same study variables (coping strategies, challenges and demographic characteristics), published separately. 

In table 2, the p-values are revised with equal decimal values. Results in section 3.3 is revised for better understanding. Figure 1 was basically the summary of the results and was based on theoretical framework that we wanted to present. As per your suggestion, we have removed it from results and placed it in the discussion. 

Point4: Discussion: The first sentence of the discussion should summarize the key results obtained in the study, and not present a description of the study and general assumptions. Apart from the comment that the study cannot be generalized due to the sample size(?), no limitations of the conducted study were presented. I believe that the listed items are major omissions that must be corrected.

Response4: Yes, we agree that the first line of the discussion should summarize the key results, we have done that. We have listed more limitations of the study as suggested. 

Point5: Conclusion: Too long, unclear and does not present a concrete conclusion of the study.

Response5: Yes, the conclusion needed to be revised and we have done that as per your suggestion. 

Point6: None of the references are written correctly according to the journal's guidelines (all are written in the usual Vancouver style).

Response6: The references are now as per the journal guidelines (ACS style). 

We hope that this revision is now up to your expectations. We look forward to hearing from you soon. 

Thanks

Reviewer 2 Report

Dear authors:

Thank you for allowing me to read your work before hand.

Although the theme is interesting for readers and although it seems to be a valuable and original study, in its present form is hard to read and understand. It requires a major revision in all chapters to bring better flow and understanding. It also requires extensive editing of english language and style.

Following my major comments/suggestions:

You decided to write introduction divided in sections. That doesn't help to understand the theme your trying to study. Also, the way you present your aims and hypothesis is not coherent with your methodology and your presentation of results.

The sampling method using the ration 95%-5% is explained, then in your results you don't meet the criteria you have established before and you don't explain the implications of that matter in your results.

The conclusion is not clear and in line with the presented results. I also miss the analysis of the limitations of the study and suggestions for present or future practice.

I suggest a major revision of content of text, style and language.

Best regards

Author Response

Dear reviewer, 

Greetings

We thank you for your time and effort in reviewing our manuscript. The feedback has been invaluable in improving the content and presentation of the paper. In accordance with your suggestions, we have revised our manuscript. The changes along with English editing are highlighted in the attached manuscript, and our point-by-point responses are given in "red" after the word “Response:” below-

Point1: You decided to write introduction divided in sections. That doesn't help to understand the theme your trying to study. Also, the way you present your aims and hypothesis is not coherent with your methodology and your presentation of results.

Response1: Yes, we agree that the introduction was written without clear theme and clarity. As per your suggestion, we have revised introduction and have removed the sections. The objectives and hypothesis are also revised so that they can address the methodology and results. 

Point2: The sampling method using the ratio 95%-5% is explained, then in your results you don't meet the criteria you have established before and you don't explain the implications of that matter in your results.

Response2: We agree that there was an issue with the sampling ratio mentioned and then the final sample we included in our study. We missed the clarification on the sample size somehow. We thank you for highlighting this point and we have revised the sampling methodology section. 

Point3: The conclusion is not clear and in line with the presented results. I also miss the analysis of the limitations of the study and suggestions for present or future practice.

Response3: We have revised the conclusion and added more limitations. Suggestions for future practice have been added in the limitations section point by point. 

Point4: I suggest a major revision of content of text, style and language.

Response4: We have revised the paper along with extensive English editing, we hope that this will present the paper in a clear manner. 

We hope that this revised manuscript is now up to your expectations. We look forward to hearing from you soon.

Thanks

Round 2

Reviewer 1 Report

Dear authors,

Even though certain parts of the manuscript have been significantly improved, during the change significant errors in the writing of the text remained and it is currently very difficult to read (especially P values, beginnings of fragments - such as “study population”, etc.). Table 2 now contains P values that were probably written twice by mistake (e.g. P<0.001001). "Graph 1" should be "Figure 1", and it is unclear to me why it was added in this form at all since it shows absolute values that have absolutely no significance for displaying the results (it is necessary to show relative values with percentages or at least write the total number respondents in order to be able to put the stated results in the context of the research). There is no need to state both the null hypothesis and the alternative hypothesis at all. Why was the 3rd objective that was listed in the original version of the manuscript removed?

Despite the changes made to the manuscript, due to numerous mistakes in writing, unclear and complicated presentation of the results, and overall results that cannot be generalized to a larger population (only a small area in India), unfortunately, I believe that this manuscript cannot be accepted for publication in International Journal of Environmental Research and Public Health.

Author Response

Dear reviewer, 

Thank you for reviewing our revised manuscript and providing your valuable feedback on the same. Please see below the response highlighted in "red" after the feedback point:

Point1: Even though certain parts of the manuscript have been significantly improved, during the change significant errors in the writing of the text remained and it is currently very difficult to read (especially P values, beginnings of fragments - such as “study population”, etc.). Table 2 now contains P values that were probably written twice by mistake (e.g. P<0.001001). "Graph 1" should be "Figure 1", and it is unclear to me why it was added in this form at all since it shows absolute values that have absolutely no significance for displaying the results (it is necessary to show relative values with percentages or at least write the total number respondents in order to be able to put the stated results in the context of the research). There is no need to state both the null hypothesis and the alternative hypothesis at all. Why was the 3rd objective that was listed in the original version of the manuscript removed?

Response1: The P values in table 2 are revised as they were written twice by mistakenly. The errors in text writing are revised. During the first revision, we added a graph (as graph1) to show the summary of issues and challenges because it was highlighted that the results were difficult to understand. Now after knowing that it does not add any value or significance in the paper, we have removed it. Additionally, we have added Mean and SD values in the result section as we felt those were missing as part of the descriptive analysis. We removed the 3rd objective because we added hypothesis statement in the first revision. Now, we have re-added the 3rd objective and removed the hypothesis statement as per your suggestion. 

We have now shared a revised pdf document with highlighting only the changes done during the second revision. 

For English editing that was already done during the first revision, we have submitted English editing certificate for the same to the Managing Editor. 

We look forward to hearing from you soon. 

Thanks

Reviewer 2 Report

Thank you for your effort.

Author Response

Dear reviewer, 

We thank you for your suggestions and final approval. 

Thanks